# Prediction of hospital readmission of multimorbid patients using machine learning models

Jules Le Lay[1], Edgar Alfonso-Lizarazo[2], Vincent Augusto[1‡]*, Bienvenu Bongue[3,4], Malek Masmoudi[5], Xiaolan Xie[1], Baptiste Gramont[6], Thomas Célarier[4,7,8‡]

1 Mines Saint-Etienne, Univ Clermont Auvergne, INP Clermont Auvergne, CNRS, UMR 6158 LIMOS, Centre CIS, Saint-Étienne France, 2 Université de Lyon, Univ Jean Monnet Saint-Étienne, LASPI, EA3059, Saint-Étienne, France, 3 Centre technique d'appui et de formation des centers d'examens de santé (CETAF), INSERM, U1059, SAINBIOSE, Dysfonction Vasculaire et Hémostase, Université de Lyon, Université Jean Monnet, Saint-Étienne, France, 4 Chaire Santé des Ainés, University of Jean Monnet, Saint-Étienne, France, 5 University of Sharjah, College of Engineering, Sharjah, United Arab Emirates, 6 Department of Internal Medicine, University Hospital of Saint-Etienne, Saint-Étienne, France, 7 Department of Clinical Gerontology, University Hospital of Saint-Etienne, Saint-Étienne, France, 8 Gérontopôle Auvergne-Rhône-Alpes, Saint-Étienne, France

☯ These authors contributed equally to this work.
‡ VA and TC also contributed equally to this work.
* augusto@emse.fr

**Data Availability Statement:** The data cannot be shared publicly. Data are available from the CNIL for researchers who meet the criteria for access to confidential data (data accessed with CNIL

## Abstract

### Objective

The objective of this study is twofold. First, we seek to understand the characteristics of the multimorbid population that needs hospital care by using all diagnoses information (ICD-10 codes) and two aggregated multimorbidity and frailty scores. Second, we use machine learning prediction models on these multimorbid patients characteristics to predict rehospitalization within 30 and 365 days and their length of stay.

### Methods

This study was conducted on 8 882 anonymized patients hospitalized at the University Hospital of Saint-Étienne. A descriptive statistical analysis was performed to better understand the characteristics of the patient population. Multimorbidity was measured using raw diagnoses information and two specific scores based on clusters of diagnoses: the Hospital Frailty Risk Score and the Calderon-Larrañaga index. Based on these variables different machine learning models (Decision Tree, Random forest and k-nearest Neighbors) were used to predict near future rehospitalization and length of stay (LoS).

### Results

The use of random forest algorithms yielded better performance to predict both 365 and 30 days rehospitalization and using the diagnoses ICD-10 codes directly was significantly more efficient. However, using the Calderon-Larrañaga's clusters of diagnoses can be used as an

authorization number 919300). CNIL web site: https://www.cnil.fr/.

**Funding:** This work was funded by the 'Agence Nationale de la Recherche', which funded the thesis on the study of multimorbid pathways under grant number ANR-18-CE19-0016. (thesis of author JLL, project manager VA). https://anr.fr/ The funders had no role in study design, data collection and analysis, decision to publish, or preparation of the manuscript.

**Competing interests:** The authors have declared that no competing interests exist.

efficient substitute for diagnoses information for predicting readmission. The predictive power of the algorithms is quite low on length of stay indicator.

## Conclusion

Using machine learning techniques using patients' diagnoses information and Calderon-Larrañaga's score yielded efficient results to predict hospital readmission of multimorbid patients. These methods could help improve the management of care of multimorbid patients in hospitals.

## Introduction

The management for care of multimorbid patients, in hospitals is a rising concern among the scientific community. Multimorbid patients tend to have more complex needs and require coordinated care from several providers [1]. Multimorbidity, defined as the "co-occurrence of multiple chronic or acute diseases and medical conditions within one person" [2], is highly prevalent in Europe. Based on the Survey on Health, Aging and Retirement in Europe (SHARE) Nielsen et al. [3] found that 31.42% of the participants above 50 years old in 14 European countries and Israel were affected by multimorbidity.. Southern Europe (Italy, Spain, France and Israel) and Northern Europe (Denmark, Sweden, and the Netherlands) had a slightly lower multimorbidity prevalence (29.8% and 26.2% respectively).

Currently, there are multiple research projects to improve the overall quality of care both inside and outside of healthcare centers, establishing dedicated care pathways for multimorbid patients [4–6].

Barnett et al. [7] reported an association between age, sex, deprivation and multimorbidity based on a list of 40 medical conditions. This list was built using policy recommendations and important chronic conditions identified in [8]. However, counting conditions can be quite limiting, and are a controversial measure of multimorbidity for these studies [9]. [10] highlighted the importance of using a standard measure of multimorbidity to analyze and compare the results of studies in which different scores have been built to describe multimorbidity.

The most common measure of multimorbidity is the Charlson comorbidity index score, originally introduced in 1987 [11] and first updated in 1994 [12] and numerous times since to be applied with administrative databases [13, 14] or to predict other outcomes [15]. In a recent systematic review, [9] explored the different multimorbidity measures developed outside of the counts of conditions. The hospital frailty risk score (HFRS), which uses weighted counts [16] or the Calderon-Larrañaga score, which counts clusters of diagnoses groups [17] are other ways to build a more efficient index.

Healthcare services can be monitored through several performance indicators. In this study we are interested in the patients' readmission and length of stay (LoS) indicators. Rehospitalization (or readmission) can be defined as "an admission to a hospital within a certain time frame (which can be 7, 15, 30,60, 90 days or even as long as a year) following an original (index) admission and discharge" [18]. According to [19], monitoring readmission and predicting the readmission of patients during their initial hospitalization are essential for two main reasons. First, authorities use this metric to evaluate and report the efficiency of healthcare centers, where a higher readmission count is being associated with lower efficiency. Second, providing a clinically relevant readmission risk early in a hospitalization stay allows

hospital workers to trigger preventive action and avoid subsequent admission, improving the consumption of medical supplies and the cost-effectiveness of the patients' care. This metric, from a cost-effectiveness perspective, is even more crucial for patients with additional chronic conditions as comorbidities that are associated with higher costs of care [20]. The LoS in hospitals is a key quality of care metric for patients and care providers, it relates to the occupancy rate of the service and is used to improve the care given.

Machine learning is a powerful set of data analysis techniques that identify and use patterns in data to realize predictions without explicitly specifying the procedure. Its use in healthcare over the past years has been extensive for the prediction of outcomes, as shown in [21]. A scoping review recently covered the use of machine learning algorithms for the prediction of hospital readmission [22]. According to this review there is a relatively high interest in tree-based methods (decision trees, random forest and boosted tree methods), although other techniques as neural networks and regularized logistic regression are also used. [23] predicted the LoS of stroke patients using J48 and a Bayesian network.

The objectives of the study presented in this article are 1) to describe the characteristics of elderly multimorbid patients in the studied hospital using diagnoses information, multimorbidity and frailty scores, and 2) to assess the ability of machine learning models to predict rehospitalization within 30 and 365 days and patients' length of stay, using diagnoses information and two aggregative scores: the hospital frailty risk score and the Calderon-Larrañaga score.

## Materials and methods

### Data description

The data used in the present study were extracted from the anonymized patients' electronic records of the hospital of Saint-Étienne (CHUSE) under 'Commission Nationale Informatique et Libertés' (CNIL) authorization number 919300, that also waived the need for consent [24]. An exception to the obligation to inform the patient was granted, as the effort required to contact all the patients involved in the study was deemed disproportionate. All patient data accessed during this research was anonymyzed. CHUSE is a university hospital at the heart of the regional healthcare network: *-Groupement Hospitalier de la Loire*. In 2019, CHUSE had more than one hundred thousand stays in one of the thousand beds in the medicine, surgery and obstetrics areas.

We focused on adult patients above 60 years old hospitalized in the CHUSE and discharged in 2017 with diagnoses in 2 different chapters of the ICD-10 classification. We excluded patients following a highly controlled care pathway, such as dialysis or outpatient surgery, except when this outpatient care led to an extended hospital stay.

For each stay we analyzed variables related to the general information of the patient as well as information concerning their care pathway. These variables are listed below:

- anonymous patient identifier

- age and sex of the patient

- service sequence

- date of admission in each unit

- length of stay (LoS) at each service of the sequence

- total length of stay (sum of the length of stays in individual services)

- admission modality and origin of the patient

- discharge modality and destination of the patient

- the list of diagnoses made at each service

From this list we were able to predict the same-hospital readmission within 30 and 365 days by linking the different stays of a unique patient using his/her anonymous identifier.

As previously mentioned there are numerous methods in the literature to describe multimorbidity. To compare the different approaches, we decided in this study capture multimorbidity by using all diagnoses information and using two multimorbidity scores: the hospital frailty risk score [16] and the Calderon-Larrañaga score [17].

The hospital frailty risk score [16] was developed to identify older patients presenting frailty diagnoses. A higher risk score is associated with a higher risk of adverse outcomes and a higher use of medical resources. Weights were calculated using logistic regression targetting of the identified frail population and validated for the prediction of adverse outcomes.

Calderon-Larrañaga [17] explored a different approach. To build this multimorbidity score, [17] gathered a panel of medical experts to group diagnoses into categories according to "clinical criteria and relevance (pathophysiological pathway, treatment, prognosis, and prevalence)" and defined the score of a patient as the number of categories where the patients had at least one diagnosis.

In order to capture the multimorbidity by using all diagnosis information, we built a database containing the exhaustive list of diagnoses made to the patient during their stay (3-digit ICD-10 codes) to compare the performance obtained by creating thematic groups of diagnoses.

The Charlson comorbidity index score was computed for comparative purposes using the ICD-10 translation of the index established by [13]; however, this score was not used for predictive procedures.

## Statistical analysis methodology

The use of different measures to synthesize multimorbidity in the literature and the tendency to use exhaustive data in process mining raises the question of the relevance of using aggregated scores in advanced statistical procedures. For this reason we compare the ability of several machine learning algorithms to predict readmission within 30 and 365 days and length of stay in the elderly multimorbid patient population using all diagnosis information and two multimorbidity scores: the hospital frailty risk score (HFRS) [16] and the Calderon-Larrañaga score [17]. In our study, we also use different categories of the Calderon-Larrañaga's and HFRS, to perform our statistical analysis and assess the loss of predictive power when aggregating the scores. For the Calderon-Larrañaga score we use the categories, and for HFRS we use groups of diagnoses having an equal weight in the score's calculation. We will refer to these nonaggregated versions of the scores as Calderon-Larrañaga portfolio and HFRS portfolio in the remainder of the paper. In the portfolio versions of the measures the category information was coded as a binary value equal to 1 if the category/diagnosis was active for the patient. We chose HFRS because it was designed to predict frailty and was validated for the prediction of 30-day readmission [16]. As mentioned before we built a database containing an exhaustive list of diagnoses made during the patients' stay (3-digit ICD-10 codes).

It is important to note that for the readmission analysis we excluded patients who died during their initial hospitalization, as readmission is not applicable for deceased patients. This resulted in the exclusion of 780 patients from the readmission database. However, we kept the data from these patients to predict length of stay. Palliative care or complex pathologies and care resulting in the death of the patient might be associated with longer hospital stays.

However, it is also important to note that for this study we only have access to the patients' data during their hospitalization, which means that we do not have access to their status after one year.

The machine learning methods used in this study for prediction of hospital readmission were tree classifier, a random forest classifier and a k-nearest neighbor classifier.

For the length of stay prediction we used a tree regressor and a random forest regressor. All experiments were performed using Python 3.7, pandas [25, 26] and scikit-learn [27].

For all learning experiments, the data were split between training and testing samples, with the training sample representing 75% of the original dataset. We used a grid search with cross-validation to parametrize the learning algorithms. The parameters that were tested and optimized were the depth and number of leaves for the decision tree approach, the depth, number of leaves and number of estimators for the random forest approach and the algorithm, leaf size and number of neighbors for the k-nearest neighbors approach.

As previously mentioned, we used the patients' anonymous identifiers to identify the patients stays. In addition to diagnosis information, we used the age, sex, length of stay, ED admission information and number of steps in the pathway to predict the same-hospital readmission within 30 days and 365 after discharge of the initial stay from the inpatient database. Those variables and the residential zip code (except the length of stay), were used for the prediction of length of stay.

The class size of readmitted patients within 30 days (1 965, or 15.86% of the stays) appeared to be far smaller than that of nonreadmitted patients. To address these imbalanced classes, we performed resampling techniques on the dataset using the imbalanced-learn Python package presented in [28] and used appropriate metrics to evaluate the results.

Resampling is a method that changes the composition of the dataset to allow training on a balanced dataset. There are techniques that delete samples from the majority class, others that generate samples of the minority class and some that combines the two. We used the imbalanced-learn Python package presented in [28]. The different methods were applied to train the learning algorithm on the balanced dataset, and we selected the best performing combination of resampling and learning algorithm. For the prediction of hospital readmission within 30 days we used the same 3 classifier algorithms: decision tree, random forest and k-nearest neighbor.

Some metrics such as accuracy are not an appropriate target to assess the performance of algorithms when dealing with unbalanced datasets. Therefore, we decided to focus on the F1-score, a weighted average of precision and recall, and receiving operating characteristic area under the curve (ROC-AUC) which compares sensitivity and specificity. Thus, we will have a better understanding of the classifier's efficiency.

To evaluate regression algorithms, we used two well-known metrics: mean absolute error and mean squared error. The classifiers' performances were assessed using accuracy (percentage of correctly predicted instance) and F1-score for rehospitalization within 365 days and receiver operating characteristic area under curve and F1-score for hospitalization within 30 days, where patients were unevenly distributed between positive and negative classes.

## Database

Table 1 presents the general variables used by the prediction algorithm. Additionally the algorithms use information on the diagnoses made to the patients. Five versions of this database were coded in order to compare the diagnoses information and aggregated multimorbidity scores:

1. using raw information on the diagnoses: one column per diagnoses with binary values;

**Table 1. General variables regarding the patient and descriptive information on their values.**

| Variable | Nature | Signification | Mean | Range (min-max) | Std dev |
|---|---|---|---|---|---|
| pat_id | Integer | Anonymous patient identifier | - | - | - |
| sex | Binary | Sex of the patient (1 = M) | | | |
| age | Integer | Age of the patient (in years) | 76.35 | 60–104 | 9.28 |
| duration | Integer | Total length of stay of the patient in hospital (in days) | 10.66 | 0–250 | 11.81 |
| geo | Zip Code | Residential zip code of the patient | - | - | - |
| nb_steps | Integer | Number of medical units visited during pathway | 1.68 | 1–14 | 1.02 |
| mortality[a] | Binary | True if the patient died during the index stay | 0.05 | - | - |
| ed_adm | Binary | True if the patient was admitted through the ED (Emergency department) | 0.36 | - | - |
| re_hosp30 | Binary | True if the patient was readmitted within 30 days after index hospitalization | 0.16 | - | - |
| re_hosp365 | Binary | True if the patient was readmitted within 365 days after index hospitalization | 0.47 | - | - |

[a] Predictive features for length of stay.

2. using Calderon-Larrañaga score;

3. using Calderon-Larrañaga portfolio (as explained in section Statistical Analysis Methodology);

4. using the HFRS score; and

5. using the HFRS portfolio.

These features were used to train and test prediction models targeting (i) readmission within 365 days, (ii) readmission within 30 days after the end of the initial hospitalization and (iii) the total length of stay.

## Results

### Epidemiological description

As specified above, we include patients above 60 years old with diagnoses in 2 different categories of ICD-10 classification who were discharged in 2017 in our study. This database includes 12 391 hospital stays of 8 882 unique patients (4 541 male patients and 4 341 female patients). The distribution of ages for this population is shown in Fig 1. On the left side of the graph are male patients, and on the right side of the graph are female patients.

We observe that the mean age for both male and female patients is quite high (74.73 and 77.93 respectively), with a median (74 and 79 years old, respectively). In our study, it was more likely that older multimorbid people were more susceptible to hospitalization.

We calculated the age-adjusted Charlson comorbidity index score [12], the Calderon-Larrañaga score [17] and the HFRS [16] for each patient. The mean age-adjusted Charlson's score was 5.65, the mean Calderon-Larrañaga's score was 4.94 and the mean HFRS was 6.32. The mean absolute difference between age-adjusted Charlson comorbidity and Calderon-Larrañaga scores was 2.39, which shows how the different scores highlight different multimorbidity profiles.

Table 2 shows the ten ICD-10 codes that are the most frequently diagnosed in the 12 391 stays of the database. Eight out of these 10 codes refer to chronic diseases, among them we can notice hypertension, that appears in 51.6% of the stays, type 2 diabetes mellitus (25.7%), overweight and obesity (13.4%), and chronic ischemic heart diseases (13.3%).

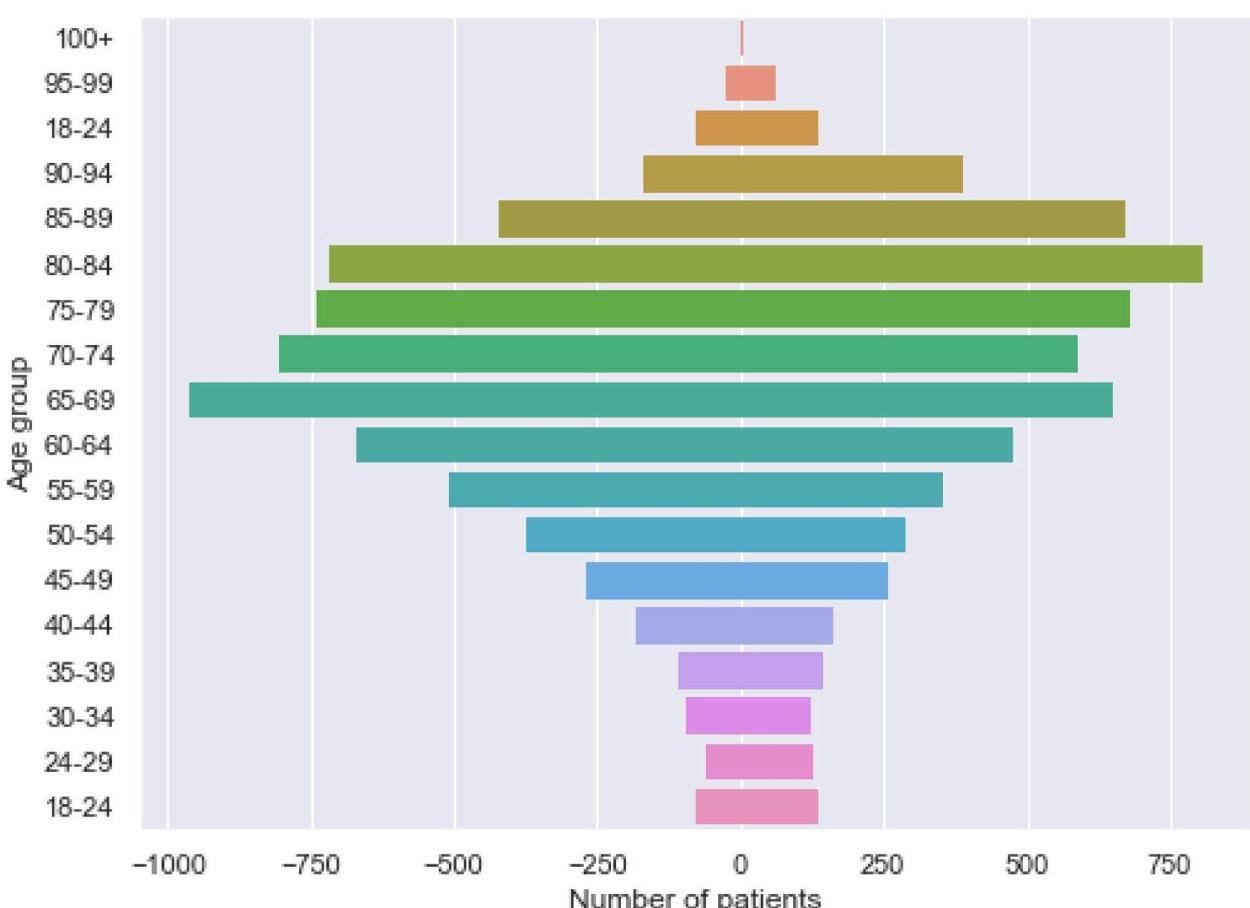

**Fig 1. Age pyramid of the multimorbid patients of the hospital of Saint-Étienne (male patients on the left side of the graph and female patients on the right side of the graph).**

**Table 2. The ten most frequent diagnoses appearing in the database.**

| ICD-10 Code | diagnoses | Number of occurrences |
|---|---|---|
| I10 | Essential hypertension | 6 391 |
| E11 | Type 2 diabetes mellitus | 3 189 |
| I48 | Atrial fibrillation | 2 681 |
| E78 | Disorders of lipoprotein metabolism [a] | 2 512 |
| N18 | Chronic kidney disease | 2 063 |
| I50 | Heart failure | 2 035 |
| J96 | Respiratory failure | 1 718 |
| E66 | Overweight and obesity | 1 658 |
| I25 | Chronic ischemic heart disease | 1 643 |
| Z74 | Problems related to care-provider dependency | 1 626 |

[a] E78: "Disorders of lipoprotein metabolism and other lipidemias", contains more specific sub-categories, such as "E780: Pure hypercholesterolemia", "E781: Pure hyperglyceridemia".

**Table 3. Accuracy, F1-score and computation times obtained for the prediction of readmission within 365 days.**

| Metric | Accuracy | F1-score | Computation time |
|---|---|---|---|
| DT[1]—All Diags | 0.597 [0.580–0.614] | 0.540 [0.518–0.563] | 557.7s |
| RF[2]—All Diags | 0.826 [0.811–0.840] | 0.812 [0.794–0.829] | 5 291.4s |
| KNN[3]—All Diags | 0.549 [0.532–0.568] | 0.551 [0.532–0.572] | 335.1s |
| DT—CL[4] score | 0.547 [0.529–0.564] | 0.527 [0.505–0.551] | 6.7s |
| RF—CL score | 0.616 [0.598–0.634] | 0.632 [0.612–0.651] | 1 472.2s |
| KNN—CL score | 0.534 [0.517–0.552] | 0.542 [0.521–0.563] | 2.8s |
| DT—CL portfolio | 0.595 [0.578–0.613] | 0.570 [0.548–0.592] | 29.5s |
| RF—CL portfolio | 0.730 [0.716–0.748] | 0.704 [0.685–0.725] | 1 754.8s |
| KNN—CL portfolio | 0.544 [0.527–0.560] | 0.551 [0.528–0.560] | 98.9s |
| DT—HFRS score | 0.553 [0.536–0.570] | 0.523 [0.502–0.545] | 8.2s |
| RF—HFRS score | 0.594 [0.577–0.611] | 0.607 [0.587–0.627] | 1 581.6s |
| KNN—HFRS score | 0.550 [0.531–0.567] | 0.568 [0.544–0.588] | 21.8s |
| DT—HFRS portfolio | 0.576 [0.559–0.594] | 0.583 [0.562–0.602] | 18.3s |
| RF—HFRS portfolio | 0.719 [0.702–0.734] | 0.696 [0.676–0.715] | 1 583.3s |
| KNN—HFRS portfolio | 0.533 [0.515–0.552] | 0.548 [0.527–0.568] | 96.3s |

[1] DT = Decision Tree,

[2] RF = Random forest,

[3] KNN = K-nearest neighbors,

[4] CL = Calderon-Larrañaga.

### Prediction of readmission within 365 days

The results obtained from the different combination of algorithm and multimorbidity evaluation are displayed in Table 3. We display the mean observed value and the 95% confidence interval calculated using a bootstrap method.

Random forest appears to be the best performing algorithm. The best performance measures, accuracy (0.826) and f1-score (0.812), are obtained by using all diagnosis information. Using the aggregated multimorbidity and frailty scores returns smaller performances. However by using the components of the scores (portfolios) the performance improves. For example using Calderon-Larrañaga's portfolio we obtain an accuracy of 0.730 and F1-score of 0.704. For HFRS with components we obtain 0.719 and 0.696 respectively with the random forest classifiers. A similar behavior is observed with KNN classifiers.

Although using all diagnostic information provides the best performance, the computational time is higher compared to the computation time of the aggregated multimorbidity scores.

A good time-performance trade-off is obtained by using Calderon-Larrañaga portfolio and random forest, as the decrease in accuracy and F1-score is small (−0.096 and −0.108, respectively), with a net improvement in the computation time. Using the HFRS portfolio and random forest is the second best option. These performance gains can be explained by the thematic groups of diagnoses created by the experts specifically for the score, which must result in easier analysis for the algorithms. Both metrics indicate that the random-forest algorithm, used on the Calderon-Larrañaga and HFRS portfolios are viable alternatives for the prediction of hospital rehospitalization within 365 days.

### Prediction of readmission within 30 days

As presented in section, we implemented resampling methods to train the algorithms on balanced training sets before testing it on unbalanced sets. We used oversampling (random

oversampling, synthetic minority oversampling technique or SMOTE and variations), under-sampling (random undersampling, near miss, condensed nearest neighbors, Tomek links, edited nearest neighbors) techniques, as well as mixed methods (SMOTE combined with edited nearest neighbors (SMOTE-ENN) or SMOTE combined with Tomek links). All related results are available in Fig 2 of the appendix *Results of the resampling techniques used to predict the within 30 day readmission.*

We singled out the most efficient combinations of classifier and resampling techniques for each dataset to perform a grid search with cross validation when considering the combination of F1-score and ROC-AUC. For the five datasets, it appeared that random undersampling was the most efficient technique.

The performance is higher when the algorithm considers all information related to the patients' diagnoses, as seen in Table 4. For example the combination of random forest and random undersampling with all diagnoses information provides an ROC-AUC score of 0.625 with a 95% Confidence Interval (CI) of [0.602–0.649] on the testing samples. Similar to the results for readmission within 365 days, using Calderon-Larrañaga portfolio is more efficient than using the aggregated score alone. With an ROC-AUC score of 0.594 [0.573–0.620] this algorithm gives the best result after all diagnoses and is significantly more efficient than the

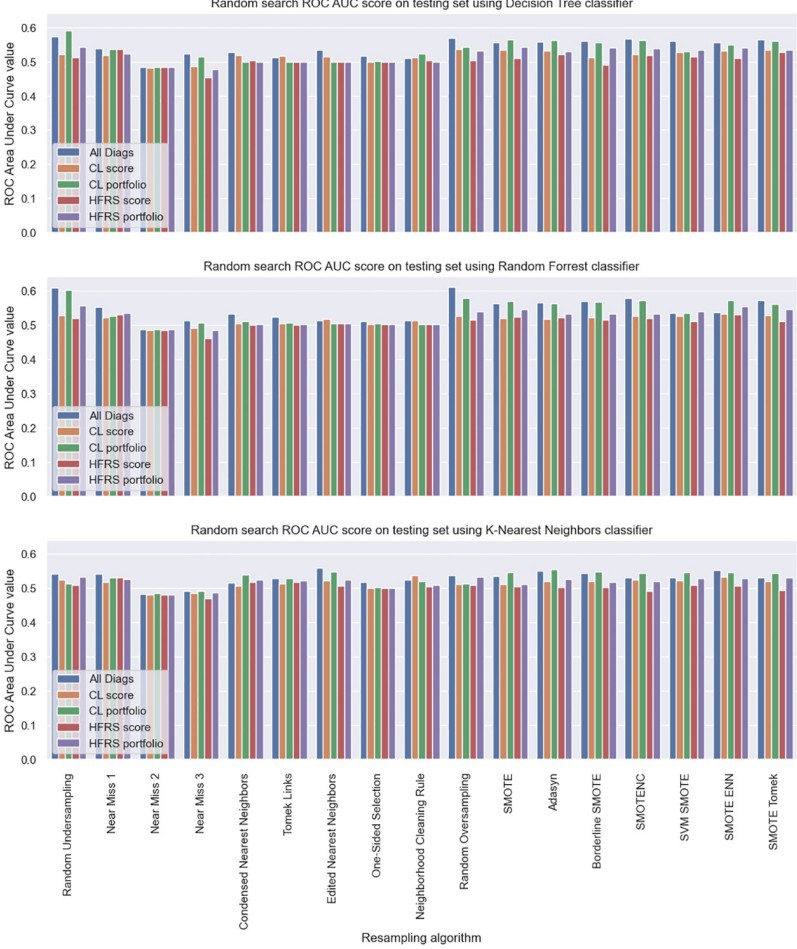

**Fig 2. Results of the different resampling techniques and classifiers for the prediction of readmission within 30 days.**

**Table 4. Accuracy, F1-score and ROC-AUC results obtained for the prediction of readmission within 30 days.**

| Best combination | Set | Accuracy | F1-score | ROC AUC |
|---|---|---|---|---|
| All diags: RF and RU[1] | Balanced TrS[2] | 0.891 [0.879–0.902] | 0.889 [0.877–0.901] | 0.891 [0.879–0.902] |
| | TrS | 0.703 [0.694–0.712] | 0.500 [0.483–0.515] | .772 [0.763–0.781] |
| | TeS[3] | 0.603 [0.586–0.620] | 0.358 [0.328–0.387] | 0.625 [0.602–0.649] |
| CL score: RF and RU | Balanced TrS | 0.636 [0.620–0.652] | 0.649 [0.632–0.667] | 0.636 [0.619–0.653] |
| | TrS | 0.544 [0.535–0.555] | 0.334 [0.319–0.349] | .596 [0.583–0.608] |
| | TeS | 0.526 [0.507–0.545] | 0.307 [0.275–0.334] | 0.565 [0.535–0.589] |
| CL portfolio: RF and RU | Balanced TrS | 0.829 [0.816–0.842] | 0.834 [0.822–0.848] | 0.829 [0.817–0.842] |
| | TrS | 0.634 [0.624–0.645] | 0.444 [0.427–0.460] | 0.725 [0.713–0.735] |
| | TeS | 0.556 [0.537–0.575] | 0.331 [0.304–0.358] | 0.594 [0.573–0.620] |
| HFRS: RF and RU | Balanced TrS | 0.633 [0.616–0.649] | 0.653 [0.634–0.670] | 0.633 [0.615–0.650] |
| | TrS | 0.535 [0.525–0.546] | 0.335 [0.320–0.349] | 0.597 [0.583–0.610] |
| | TeS | 0.510 [0.491–0.532] | 0.304 [0.275–0.328] | 0.561 [0.536–0.584] |
| HFRS' portfolio: RF and RU | Balanced TrS | 0.835 [0.821–0.850] | 0.830 [0.813–0.843] | 0.835 [0.822–0.849] |
| | TrS | 0.645 [0.644–0.664] | 0.440 [0.422–0.455] | 0.713 [0.702–0.723] |
| | TeS | 0.562 [0.544–0.580] | 0.309 [0.282–0.336] | 0.571 [0.547–0.594] |

[1] RU: Random Undersampling,

[2] TrS: Training Set,

[3] TeS: Testing Set.

experiments using HFRS. The ROC curves and calibration curves are displayed in Fig 3. We note that the results are quite different for HFRS, and the ROC-AUC and F1-score are both comparable for the two versions of this multimorbidity index (the HFRS score and HFRS portfolio).

## Prediction of length of stay

Overall, the random forest algorithm performed better than the decision tree, with an improvement of 10 to 30 square days in mean squared error from the decision tree results. This improvement is the most significant when using all available diagnoses (-0.802 MAE and -30.462 MSE). These results can not be considered conclusive as the best mean absolute error is still above 5 days, which represents half of the mean length of stay in the database. The raw results are displayed in Table 5.

## Discussion

In this study we included patients based on the number of diagnoses in different ICD-10 categories, before applying different scores (the Charlson comorbidity score, Calderon-Larrañaga score and HFRS).

The mean age of the studied patients is 76 years. In addition, female multimorbid patients tend to be older than male patients. The mean age-adjusted Charlson morbidity score was 5.65, and the mean Calderon-Larrañaga score was 4.94. The most prevalent diseases diagnosed in patients were hypertension and type 2 diabetes mellitus.

We built 5 versions of our database for each outcome, each taking into account various information on the diagnosis. The raw diagnosis information was compared to 2 aggregation scores: the HFRS, a score built to measure the frailty of patients based on diagnoses information, and the Calderon-Larrañaga score, a multimorbidity score based on groups of diagnoses.

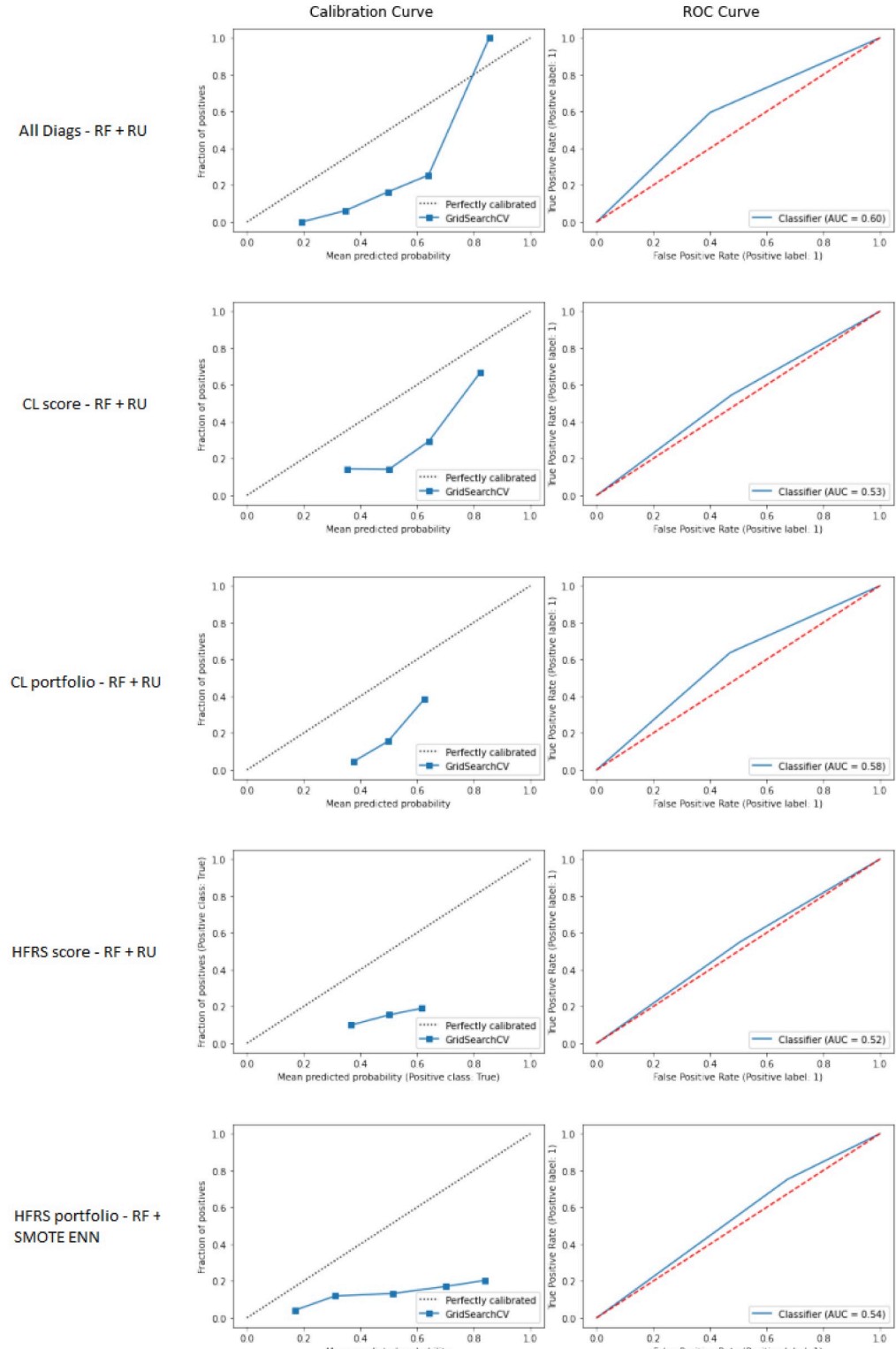

**Fig 3. Results of the different resampling techniques and classifiers for the prediction of readmission within 30 days.**

**Table 5. Length of stay prediction results.**

| Algorithm | MAE | MSE |
|---|---|---|
| Decision Tree—All diags | 6.010 | 103.149 |
| Random forest—All diags | 5.208 | 72.687 |
| Decision Tree—Calderon-Larrañaga's score | 6.297 | 97.854 |
| Random forest—Calderon-Larrañaga's score | 6.146 | 88.163 |
| Decision Tree—Calderon-Larrañaga's portfolio | 5.911 | 82.767 |
| Random forest—Calderon-Larrañaga's portfolio | 5.894 | 81.903 |
| Decision Tree—HFRS score | 5.849 | 104.728 |
| Random forest—HFRS score | 5.609 | 77.532 |
| Decision Tree—HFRS portfolio | 6.137 | 100.737 |
| Random forest—HFRS portfolio | 5.728 | 82.811 |

The prediction of medium-term readmission was efficient, with the best score achieved using random forest and all diagnoses information (an accuracy score of 0.826 [0.811–0.840] and a F1-score of 0.812 [0.794–0.829]). Using Calderon-Larrañaga portfolio resulted in a slight decrease in performance for both indicators (−0.096 and −0.108 respectively). We believe that the clusters of diseases used in Calderon-Larrañaga portfolio can be used as an efficient substitute of diagnoses information for predicting readmission within 365 days after initial discharge.

We tested multiple resampling solutions to account for the imbalance in the rehospitalization within 30 days dataset. We obtained at best an ROC-AUC score of 0.625, which is acceptable, although it is slightly lower than the recent results in the literature, as [22] reported a median AUC of 0.68 on the studies they identified. The accuracy on the unbalanced testing set is of 0.603. When using the combination of Calderon-Larrañaga portfolio, random forest and random undersampling, we obtained a mean ROC-AUC score of 0.594 and a mean accuracy of 0.556. The use of Calderon-Larrañaga portfolio can be a viable alternative for the prediction of the within 30 day readmission on a medico-administrative database.

Overall, the predictive power of our algorithms is quite low for length of stay prediction. The use of a random forest regressor gave the best results for the two metrics used, MAE and MSE. Both the HFRS and the Calderon-Larrañaga score were outperformed by the use of all information on diagnosis. However, the HFRS performed better than the Calderon-Larrañaga score. The HFRS seems to be an acceptable alternative to the use of exhaustive information on diagnosis with a random forest algorithm for predicting length of stay.

In general, the experiments show that for readmission within 365 days, using all diagnosis gives the best results. The Calderon-Larrañaga and HFRS give comparable results. Calderon-Larrañaga portfolio gives significantly better results than the experiments using HFRS scores and portfolio for the prediction of readmission within 30 days, but is still outperformed by the random forest with a random undersampling on all diagnosis information. The different experiments show that the machine learning algorithms for the prediction of length of stay give at best a mean absolute error of 5.208 and mean square error of 72.687 (random forest used with HFRS score).

Calderon-Larrañaga portfolio is a standardized and thorough tool that accounts for most ICD-10 codes and chronic conditions, and we believe that it gives a quite accurate view on multimorbidity. Thus, it can be a good alternative to using the raw information on diagnoses for predicting readmission within 365 and 30 days.

The main limitations of this study are related to the lack of information on the vital status of patients after their hospitalization. This could represent a bias, as it is possible that patients

died between the initial discharge and the end of the period of interest. Thus, the absence of a hospital stay within 30 or 365 days after discharge may be caused by the death of the patient.

A key component of multimorbidity according to Barnett et al. [7] that we could not grasp in this study is the socioeconomic aspect. Similar to [7], we can use geographical information as a proxy, but we do not have access in the present database to an evaluation of the socioeconomic status per area. In addition, we have access only to the residential zip-code of the patient, which can cover quite a large area and hide many disparities between patients. A favorable familial situation, with an available caregiver, is a key component of the support and recovery of patients, and this information is not available to us. We believe that these two key components of care for multimorbid patients would be a valuable addition when using the methodology presented in this paper.

## Conclusion

In this study we described the general characteristics of the multimorbid population hospitalized in the CHUSE in 2017 using diagnoses information, multimorbidity or frailty scores and various machine learning techniques were used to predict key components of the hospitalization pathway: the length of stay and the rehospitalization within 30 and 365 days after initial discharge.

Random forrest algorithms were the most efficient for predicting those three outcomes. Using all diagnoses information gave better results at the price of high calculation times. Using the Calderon-Larrañaga portfolio is an efficient alternative for rehospitalization within 30 and 365 days. However, for LoS prediction, the HFRS metrics gave the best results.

For future research, we intend to apply prediction techniques on patient data to single out complicated pathways and combine this with the results obtained in this study in a discrete event simulation model. Our goal is to evaluate the organizational impact of redirecting those patients toward a newly created unit, dedicated to multimorbid patients. It would also be of interest to combine the approach of this paper with prospective data at admission, which would provide additional valuable information, such as socioeconomic data and familial context, respecting the anonymity of patients. This could allow us to evaluate if the patient would be fit for the new multimorbid pathway and to track the decision process of the care team when deciding in which unit the patient must be routed.

## Appendices: Results of the resampling techniques used to predict the within 30 day readmission

Fig 2 shows the results obtained when testing the different resampling techniques. The algorithms were trained on a dataset balanced using the specified resampling technique using a random search with cross validation from the scikit-learn package [27]. Only the ROC-AUC score on the testing set is displayed here.

## Appendices: ROC and calibration curves for the prediction of readmission within 30 days

Fig 3 displays the ROC and calibration curves for the 5 best performing algorithms for predicting 30 days all-cause readmission. We used the scikit-learn package [27] functions to generate those curves. They were generated using the testing set.

## Author Contributions

**Data curation:** Baptiste Gramont, Thomas Célarier.

**Formal analysis:** Jules Le Lay.

**Funding acquisition:** Vincent Augusto.

**Methodology:** Edgar Alfonso-Lizarazo, Vincent Augusto.

**Project administration:** Xiaolan Xie.

**Software:** Jules Le Lay.

**Supervision:** Malek Masmoudi, Xiaolan Xie.

**Validation:** Bienvenu Bongue, Baptiste Gramont, Thomas Célarier.

**Writing – original draft:** Jules Le Lay.

**Writing – review & editing:** Edgar Alfonso-Lizarazo, Bienvenu Bongue, Malek Masmoudi.

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
