## [Decision Letter · Decision Letter 0]

20 Apr 2022

PONE-D-22-08556Prediction of Hospital Readmission of Multimorbid Patients Using Machine Learning ModelsPLOS ONE

Dear Dr. LE LAY,

Thank you for submitting your manuscript to PLOS ONE. After careful consideration, we feel that it has merit but does not fully meet PLOS ONE’s publication criteria as it currently stands. Therefore, we invite you to submit a revised version of the manuscript that addresses the points raised during the review process.

We look forward to receiving your revised manuscript.

Kind regards,

Antonio De Vincentis

Academic Editor

PLOS ONE

Journal Requirements:

3. Please ensure you include in the Methods section of your manuscript all information regarding the data access authorization you have provided in the Ethics Statement section of the online submission form, including information regarding the exception for patient consent granted by the authority who approved the data access.

Additionally, please note that PLOS ONE has specific guidelines on code sharing for submissions in which author-generated code underpins the findings in the manuscript. In these cases, all author-generated code must be made available without restrictions upon publication of the work. Please review our guidelines at https://journals.plos.org/plosone/s/materials-and-software-sharing#loc-sharing-code and ensure that your code is shared in a way that follows best practice and facilitates reproducibility and reuse.

Reviewers' comments:

Reviewer's Responses to Questions

**Comments to the Author**

1. Is the manuscript technically sound, and do the data support the conclusions?

Reviewer #1: Yes

Reviewer #2: Yes

Reviewer #3: Partly

2. Has the statistical analysis been performed appropriately and rigorously? 

Reviewer #1: Yes

Reviewer #2: Yes

Reviewer #3: I Don't Know

3. Have the authors made all data underlying the findings in their manuscript fully available?

Reviewer #1: Yes

Reviewer #2: No

Reviewer #3: No

4. Is the manuscript presented in an intelligible fashion and written in standard English?

Reviewer #1: Yes

Reviewer #2: Yes

Reviewer #3: Yes

5. Review Comments to the Author

Reviewer #1: I would like to thank the authors for their work.

This is an interesting paper, which aims to understand the characteristics of the multimorbid population that needs hospital care by using all diagnoses information and two aggregated multimorbidity and frailty scores, and to use machine learning prediction models on these multimorbid patients characteristics to predict rehospitalization within 30 and 365 days and their length of stay.

Before the publication I would just highlight two possible minor revisions:

1) Page 9, line 31: the quotation mark before "an admissione" seems "inverted";

2) Materials and methods: in the "epidemiological description" section some results are described. I would suggest to adapt and move this part to the "Results" section.

Reviewer #2: The Authors present an interesting study, aiming at understanding the characteristics of the multimorbid population that needs hospital care and predict, using ML algorithms, the re-hospitalization.

The paper is generally well written.

My comments:

- I think that, for the models listed in tab 4, a figure showing the ROC curves may be beneficial for the readers as it would allow to understand if cut-offs other than predict probability = 0.5 are viable and what is their performance. This may be particularly interesting for the potential “clinical” application of these models where, according to one’s aim, a trade-off between sensitivity and specificity may be acceptable.

- I think that for the models listed in tab 4, reliability plots should be shown. This would allow the readers to better understand the reliability of the proposed models. It may also help the Authors to understand if any problem with calibration arises. RF models, for example, may need some kind of recalibration, and this issue may be particularly important given that the models were trained on “artificial data” with solved class imbalance and applied to “real” testing data. I’d suggest the Authors to try and see if Platt scaling or isotonic regression are viable ways to improve the calibration of the models (if any issue arise).

- In general, models seem to be prone to overfitting (as expected). I would suggest to try and run the models including only a subsample of available variables (based on the importance of each variable, obtained after the fitting of complete RF models – for example best 5%, best 25%, best 50%). This may have several advantages: it may reduce the overfitting, may reduce the calculation time, and may have sense from a clinical point of view. Indeed, it has been proven that some chronic conditions (or group of conditions) are strongly linked to the risk of hospitalization (heart failure, COPD, dementia for example), whereas other are not (osteoarthritis for example). These latter conditions may even introduce some form of statistical noise in the models.

- Please, add 95%CI to the performance metrics listed in the paper (accuracy, F1, AUCs) to allow a better comparison between models.

- I’d like to know what is the performance (Accuracy, F1, AUC) of the simple count of chronic conditions according to the Calderon-Larranaga’s list of chronic diseases categories (for example, used in a simple logistic regression). This also goes for age used as single predictor. From an application point of view, it is important to know what the actual benefit of the implementation of ML models in comparison with much more simpler measures is.

- For the prediction of LoS, the performance shown by the models is not optimal (as highlighted by the Authors). Here, I would suggest, as sensitivity analyses, to either exclude or aggregate those participants with very long LoS (for example, outliers with LoS > Q75 + 2*IQR). From a clinical point of view, these very long LoS are likely to be associated with hospitalizations characterized by adverse events and complications (delirium, nosocomial infections and so on).

- The Calderon-Larranaga list of chronic conditions has been created using data from a population-based study enrolling only persons aged 60+. Has the HFRS been validated in younger persons? If not, it would be preferable to exclude patients younger than 60 from the analysis as these persons are more likely to be affected by a few chronic conditions with a strong impact on health, whereas those older are more likely to accumulate an elevated number chronic conditions.

- In some Countries, the diagnoses reported on the discharge letter from hospitals are used to obtain some kind of public (re)fundings. This is likely to introduce a bias in the reported diagnoses: those diagnoses that lead to higher (re)fundings are more likely to be reported. Is this the case also for the setting used by the Authors?

- It would be interesting to look at the predictive performance of the models in different age strata of the population. It is likely that among very old individuals with multiple chronic conditions (and likely affected by frailty and disability), a general tendency toward non-hospitalization and at-home (or nursing home) management is present. This may lead to a paradox where younger persons with multimorbidity are likely to be (re)hospitalized, whereas older persons are likely to be (re)hospitalized only when they are “healthier”.

Reviewer #3: In their paper, Le Lay et al studied the accuracy of machine learning models in predicting length of stay and hospital readmission. Despite the topic is interesting, all the sections of the study should be improved to clarify the objectives and the results of the study.

Introduction

The Introduction should be improved: the authors should explain the rationale of using machine learning to predict hospitalizations and length of stay, expand what it is known on this topic and what this study will add to the literature. Furthermore, Introduction should include the objectives of the study, that are lacking. Reading the Abstract, it is not clear which is the objective of the study. To compare different machine learning predicting models?

Methods

Epidemiological description included results such as the total number of the participants and their mean age. This information should be moved in the Results.

There is no information about ethic committee approval.

Why using three different methods to classify multimorbidity?

Which were the variables included in the learning experiments?

Results

General characteristics of the study sample should be reported. Otherwise, it is very difficult to interpret the results.

Prediction of readmission within 30 days: I think that could be more appropriate to describe resampling, the metrics used, etc in the Methods section and to report in this subsection (and in the Results section in general) only the results.

Discussion/Conclusions

The first paragraph of the discussion should report a summary of the main results of the study.

It seems that the conclusions simply discuss the results of the study (thus, most part of this section should be moved to the Discussion, comparing them with the available literature).

I suggest to report all the limitations of the study in a single paragraph and not throughout the Discussion.

The conclusion should briefly summarize the conclusions of the study, potential clinical implications and future perspectives.

In the conclusions the authors stated that they reported the general characteristics of the multimorbidity population, but I can’t find this information.

---

## [Author Response · Author response to Decision Letter 0]

9 Aug 2022

Jules Le Lay

École des Mines de Saint-Étienne

158 cours Fauriel, 

42100 Saint-Étienne

Dear Editors of the PLOS ONE journal, 

Thanks for the opportunity to submit a revised version of our manuscript entitled « Prediction of Hospital Readmission of Multimorbid Patients Using Machine Learning Models» by Le Lay, Alfonso-Lizarazo, Augusto, Bongue, Masmoudi, Xie, Gramont, and Célarier. We appreciate the efforts you and the rewiewers made to comment our manuscript, which have allowed us to improve the article.

Here under is a thorough response to the editors’ and reviewers’ comments : 

Editors: 

1) Please ensure that your manuscript meets PLOS ONE's style requirements, including those for file naming. The PLOS ONE style templates can be found at :

R/ We added elements and renamed the files to meet the requirements. Please let us know if we missed any of the points listed.

2) We suggest you thoroughly copyedit your manuscript for language usage, spelling, and grammar. If you do not know anyone who can help you do this, you may wish to consider employing a professional scientific editing service

R/ The initial manuscript was edited for language, spelling and grammar by AJE editing services. We’ll attach the editing certificate to the resubmission. The modifications in this version concern mostly the figures and tables, and only a few sentences. Please let us know if another editing is required. 

3) Please ensure you include in the Methods section of your manuscript all information regarding the data access authorization you have provided in the Ethics Statement section of the online submission form, including information regarding the exception for patient consent granted by the authority who approved the data access.

R/ We added in the Methods section the information regarding data access, this study was approved by the French national commission on informatics and liberty (CNIL) for the access to data under authorization number 919300. (This information was added to the revised paper and was already present on the online submission form). 

4) We note that you have indicated that data from this study are available upon request. PLOS only allows data to be available upon request if there are legal or ethical restrictions on sharing data publicly.

R/ Indeed, the data used for the experiments presented in the paper are anonymized patients data extracted from medical records of the University hospital of Saint-Étienne. We were given access to this database by the 'Commission Nationale Informatique et Libertés' under authorization number 919300. The authors do not have the right to share this information publicly. 

Reviewer 1: 

5) Page 9, line 31: the quotation mark before "an admissione" seems "inverted";

R/ Indeed, I did not use the right quotation mark, this was corrected in the revised version of the paper.

6) Materials and methods: in the "epidemiological description" section some results are described. I would suggest to adapt and move this part to the "Results" section.

R/ We moved this paragraph at the beginning of the “Results” section as suggested. 

Reviewer 2: 

7) I think that, for the models listed in tab 4, a figure showing the ROC curves may be beneficial for the readers as it would allow to understand if cut-offs other than predict probability = 0.5 are viable and what is their performance. This may be particularly interesting for the potential “clinical” application of these models where, according to one’s aim, a trade-off between sensitivity and specificity may be acceptable.

R/ The ROC curves have been built for those models and have been included in the revised paper.

8) I think that for the models listed in tab 4, reliability plots should be shown. This would allow the readers to better understand the reliability of the proposed models. It may also help the Authors to understand if any problem with calibration arises. RF models, for example, may need some kind of recalibration, and this issue may be particularly important given that the models were trained on “artificial data” with solved class imbalance and applied to “real” testing data. I’d suggest the Authors to try and see if Platt scaling or isotonic regression are viable ways to improve the calibration of the models (if any issue arise).

R/ Calibration curves were plotted for those models and have been included in the revised paper, in the appendix.

9) In general, models seem to be prone to overfitting (as expected). I would suggest to try and run the models including only a subsample of available variables (based on the importance of each variable, obtained after the fitting of complete RF models – for example best 5%, best 25%, best 50%). This may have several advantages: it may reduce the overfitting, may reduce the calculation time, and may have sense from a clinical point of view. Indeed, it has been proven that some chronic conditions (or group of conditions) are strongly linked to the risk of hospitalization (heart failure, COPD, dementia for example), whereas other are not (osteoarthritis for example). These latter conditions may even introduce some form of statistical noise in the models.

R/ We think that this is an interesting improvement opportunity. However, unfortunately, we did not have enough time to develop the experimentation of this approach.

10) Please, add 95% CI to the performance metrics listed in the paper (accuracy, F1, AUCs) to allow a better comparison between models.

R/ We implemented a bootstrap method to calculate the 95% CI for the metrics used in the classification metrics. This information was included in the revised paper. 

11) I’d like to know what is the performance (Accuracy, F1, AUC) of the simple count of chronic conditions according to the Calderon-Larrañaga’s list of chronic diseases categories (for example, used in a simple logistic regression). This also goes for age used as single predictor. From an application point of view, it is important to know what the actual benefit of the implementation of ML models in comparison with much more simpler measures is.

R/ Our work focuses on the use of classical machine learning methods on medico-economic datasets enriched with medical tools used in the description of multimorbid patients. In this way, although the comparison with other methods is not part of our objectives, we carried out an experiment using logistic regression for hospital readmission within 30 days with random undersampling using all diagnosis: and obtained the following results (which have not been included in the article) 

 Accuracy F1-score ROC-AUC SCORE

Balanced Training set 0.686 [0.670 – 0.704] 0.680 [0.661 – 0.698] 0.686 [0.670 – 0.703]

Training set 0.636 [0.626 – 0.645] 0.371 [0.636 – 0.663] 0.649 [0.636 – 0.663]

Testing set 0.600 [0.585 – 0.617] 0.291 [0.262 – 0.322] 0.577 [0.550 – 0.603]

Based on these results, the logistic regression with random undersampling using all diagnosis provides lower performances with respect to the method presented in the article.

12) For the prediction of LoS, the performance shown by the models is not optimal (as highlighted by the Authors). Here, I would suggest, as sensitivity analyses, to either exclude or aggregate those participants with very long LoS (for example, outliers with LoS > Q75 + 2*IQR). From a clinical point of view, these very long LoS are likely to be associated with hospitalizations characterized by adverse events and complications (delirium, nosocomial infections and so on).

R/ Although in our dataset a very small number of patients have a very long LoS compared to the average patient, we decided not to exclude or aggregate records. Indeed, we believe it is important to include these records in the training of machine learning algorithms to better identify characteristic patterns of prolonged stays. 

13) The Calderon-Larrañaga list of chronic conditions has been created using data from a population-based study enrolling only persons aged 60+. Has the HFRS been validated in younger persons? If not, it would be preferable to exclude patients younger than 60 from the analysis as these persons are more likely to be affected by a few chronic conditions with a strong impact on health, whereas those older are more likely to accumulate an elevated number chronic conditions.

R/ We have taken this remark into account and for this revised paper we have carried out the experimentation excluding patients younger than 60 and reported them in the revised paper. The results, in comparison to those taking into account the entire population (including patients younger than 60), are slightly lower or slightly higher depending on the indicator under analysis. For example, the mean absolute error and mean square error were improved for LoS prediction. Accuracy and F1-score were approximately the same for 365 days hospital readmission, with a slight improvement in the best-case scenario (using all diagnosis and random forest). Accuracy, F1-score and ROC-AUC for 30-days readmission have slightly decreased. 

14) In some Countries, the diagnoses reported on the discharge letter from hospitals are used to obtain some kind of public (re)fundings. This is likely to introduce a bias in the reported diagnoses: those diagnoses that lead to higher (re)fundings are more likely to be reported. Is this the case also for the setting used by the Authors?

R/ The anonymized patient data used in our study is extracted from the hospital electronic records. The methodology used in this study does not allow us to identify the existence of this kind of bias or to quantify it if it exists.

15) It would be interesting to look at the predictive performance of the models in different age strata of the population. It is likely that among very old individuals with multiple chronic conditions (and likely affected by frailty and disability), a general tendency toward non-hospitalization and at-home (or nursing home) management is present. This may lead to a paradox where younger persons with multimorbidity are likely to be (re)hospitalized, whereas older persons are likely to be (re)hospitalized only when they are “healthier”.

R/ As mentioned in the response to comment 13, in the revised paper we have presented the results excluding patients younger than 60, for this reason we have decided not to perform an analysis by age strata.

Reviewer 3: 

1) The Introduction should be improved: the authors should explain the rationale of using machine learning to predict hospitalizations and length of stay, expand what it is known on this topic and what this study will add to the literature. Furthermore, Introduction should include the objectives of the study, that are lacking. Reading the Abstract, it is not clear which is the objective of the study. To compare different machine learning predicting models.

R/ The introduction section has been modified including these elements. 

The objective of this study is the prediction of rehospitalization within 30 and 365 days and length of stay of multimorbid patients using machine learning models in which multimorbidity is measured using all diagnoses information (ICD-10 codes) and two aggregated multimorbidity and frailty scores. 

2) Epidemiological description included results such as the total number of the participants and their mean age. This information should be moved in the Results.

R/ We moved this section at the beginning of the “Results” section.

3) There is no information about ethic committee approval.

R/ The study was approved by the French national commission on informatics and liberty (CNIL) for the access to data under authorization number 919300. This information was added to the revised paper.

4) Why using three different methods to classify multimorbidity? 

R/ Since our study focuses on multimorbidity patients, it is important for us to consider different approaches reported in the literature to measure multimorbidity and evaluate which of them allow a better prediction of re-hospitalization for this type of patients.

5) Which were the variables included in the learning experiments?

R/ We have included in the revised paper (section materials and methods) an explicit list of the variables used in the models.

6) General characteristics of the study sample should be reported. Otherwise, it is very difficult to interpret the results. 

R/ We reported in the paper the general characteristics of the dataset in section “Results/Epidemiological description” the epidemiological features. 

7) Prediction of readmission within 30 days: I think that could be more appropriate to describe resampling, the metrics used, etc in the Methods section and to report in this subsection (and in the Results section in general) only the results.

R/ This comment has been considered in the revised paper.

8) The first paragraph of the discussion should report a summary of the main results of the study.

R/ This comment has been considered in the revised paper.

9) It seems that the conclusions simply discuss the results of the study (thus, most part of this section should be moved to the Discussion, comparing them with the available literature).

R/ This comment has been considered in the revised paper.

10) I suggest to report all the limitations of the study in a single paragraph and not throughout the Discussion.

R/ This comment has been considered in the revised paper.

11) The conclusion should briefly summarize the conclusions of the study, potential clinical implications and future perspectives.

R/ This comment has been considered in the revised paper.

12) In the conclusions the authors stated that they reported the general characteristics of the multimorbidity population, but I can’t find this information.

R/ The general characteristics of the multimorbidity population studied are described in the “Results” section, in the “Epidemiological description” subsection.

We would like to thank all reviewers for their detailed comments, we believe the modifications made to the experimental design and reporting of the results improves the quality of the paper. 

We look forward to hearing from you regarding our submission and to respond to any additional questions or comments you may have. 

Jules Le Lay, Edgar Alfonso-Lizarazo, Vincent Augusto1, Bienvenu Bongue, Malek Masmoudi, Xiaolan Xie, Baptiste Gramont and Thomas Célarier.

---

## [Decision Letter · Decision Letter 1]

26 Sep 2022

PONE-D-22-08556R1Prediction of Hospital Readmission of Multimorbid Patients Using Machine Learning ModelsPLOS ONE

Dear Dr. LE LAY,

Thank you for submitting your manuscript to PLOS ONE. After careful consideration, we feel that it has merit but does not fully meet PLOS ONE’s publication criteria as it currently stands. Therefore, we invite you to submit a revised version of the manuscript that addresses the points raised during the review process. In particular, please address the comments raised by reviewer 3.

We look forward to receiving your revised manuscript.

Kind regards,

Antonio De Vincentis

Academic Editor

PLOS ONE

Journal Requirements:

Reviewers' comments:

Reviewer's Responses to Questions

**Comments to the Author**

1. If the authors have adequately addressed your comments raised in a previous round of review and you feel that this manuscript is now acceptable for publication, you may indicate that here to bypass the “Comments to the Author” section, enter your conflict of interest statement in the “Confidential to Editor” section, and submit your "Accept" recommendation.

Reviewer #2: All comments have been addressed

Reviewer #3: (No Response)

2. Is the manuscript technically sound, and do the data support the conclusions?

Reviewer #2: Yes

Reviewer #3: Yes

3. Has the statistical analysis been performed appropriately and rigorously? 

Reviewer #2: Yes

Reviewer #3: Yes

4. Have the authors made all data underlying the findings in their manuscript fully available?

Reviewer #2: (No Response)

Reviewer #3: Yes

5. Is the manuscript presented in an intelligible fashion and written in standard English?

Reviewer #2: Yes

Reviewer #3: Yes

6. Review Comments to the Author

Reviewer #2: The revised version of the manuscript is improved and all my comments have been addressed.

Thank you!

Reviewer #3: The manuscript is significantly improved after revision. However, I have two other comments to further improve it:

The Introduction lacks of the objectives of the study (that have been added in the abstract only). Please add them.

The Conclusion section is more than one page long, reporting also the results of the study including a discussion. I suggest to shortened them (moving the discussion of the results in the Discussion), reporting in this section only the conclusions and future perspectives/clinical application.

7. PLOS authors have the option to publish the peer review history of their article (what does this mean?). If published, this will include your full peer review and any attached files.

Reviewer #2: No

Reviewer #3: **Yes: **Diana Lelli

---

## [Author Response · Author response to Decision Letter 1]

9 Nov 2022

Jules Le Lay

École des Mines de Saint-Étienne

158 cours Fauriel, 

42100 Saint-Étienne

Dear Editors of the PLOS ONE journal, 

Thank you once again for giving us the opportunity to submit a revised version of our manuscript, titled “Prediction of Hospital Readmission of Multimorbid Patients Using Machine Learning Models” to your journal. This article was written by Le Lay, Alfonso-Lizarazo, Augusto, Bongue, Masmoudi, Xie, Gramont, and Célarier. We would like to thank you and the reviewers for taking the time to review and comment this paper. 

You will find here after our responses to your comments and to the reviewers comments. 

Editor’s comments: 

Response: Thank you for this remark. We reviewed our paper carefully and checked all the references. 

We did not change the reference list, except for the addition of a DOI in reference 15, (linking to a correction of supplementary materials).

Comments from Reviewer 2: 

The revised version of the manuscript is improved and all my comments have been addressed. Thank you!

Response: Thank you very much for taking the time to review our manuscript, all the comments received in the reviewing process have helped us a lot. 

Comments from Reviewer 3:

The manuscript is significantly improved after revision. However, I have two other comments to further improve it:

1/ The Introduction lacks of the objectives of the study (that have been added in the abstract only). Please add them.

Response: Thank you for this remark, we added the objectives of the study at the end of the Introduction section, at lines 54-59: 

“The objectives of the study presented in this article are 1) to describe the characteristics of elderly multimorbid patients in the studied hospital using diagnoses information, multimorbidity and frailty scores, and 2) to assess the ability of machine learning models to predict rehospitalization within 30 and 365 days and patients’ length of stay, using diagnoses information and two aggregative scores : the hospital frailty risk score and the Calderon-Larrañaga score.”

2/ The Conclusion section is more than one page long, reporting also the results of the study including a discussion. I suggest to shortened them (moving the discussion of the results in the Discussion), reporting in this section only the conclusions and future perspectives/clinical application.

Response: The discussion and conclusion sections have been modified to improve the coherence of the paper, on pages 14-17. Comments on the results were moved to the discussion section as suggested, and we modified the structure of this section. 

Thank you once again for your comments which have allowed us to considerably improve our paper.

Dr Jules Le Lay

---

## [Decision Letter · Decision Letter 2]

7 Dec 2022

Prediction of Hospital Readmission of Multimorbid Patients Using Machine Learning Models

PONE-D-22-08556R2

Dear Dr. LE LAY,

We’re pleased to inform you that your manuscript has been judged scientifically suitable for publication and will be formally accepted for publication once it meets all outstanding technical requirements.

Kind regards,

Antonio De Vincentis

Academic Editor

PLOS ONE

Additional Editor Comments (optional):

Reviewers' comments:

Reviewer's Responses to Questions

**Comments to the Author**

1. If the authors have adequately addressed your comments raised in a previous round of review and you feel that this manuscript is now acceptable for publication, you may indicate that here to bypass the “Comments to the Author” section, enter your conflict of interest statement in the “Confidential to Editor” section, and submit your "Accept" recommendation.

Reviewer #3: All comments have been addressed

2. Is the manuscript technically sound, and do the data support the conclusions?

Reviewer #3: Yes

3. Has the statistical analysis been performed appropriately and rigorously? 

Reviewer #3: Yes

4. Have the authors made all data underlying the findings in their manuscript fully available?

Reviewer #3: No

5. Is the manuscript presented in an intelligible fashion and written in standard English?

Reviewer #3: Yes

6. Review Comments to the Author

Reviewer #3: The authors have addressed all my comments. The manuscript has significantly improved. I have no further comments.

7. PLOS authors have the option to publish the peer review history of their article (what does this mean?). If published, this will include your full peer review and any attached files.

Reviewer #3: **Yes: **Diana Lelli

---

## [Editor Report · Acceptance letter]

14 Dec 2022

PONE-D-22-08556R2 

Prediction of Hospital Readmission of Multimorbid Patients Using Machine Learning Models 

Dear Dr. Le Lay:

I'm pleased to inform you that your manuscript has been deemed suitable for publication in PLOS ONE. Congratulations! Your manuscript is now with our production department. 

Kind regards, 

on behalf of

Dr. Antonio De Vincentis 

Academic Editor

PLOS ONE